# OPEN-VOCABULARY OBJECT DETECTION FOR INCOMPARABLE SPACES

## ABSTRACT

In open-vocabulary object detection (OVDet), specifying the object of interest at inference time opens up powerful possibilities, allowing users to define new categories without retraining the model. These objects can be identified through text descriptions, image examples, or a combination of both. However, visual and textual data, while complementary, encode different data types, making direct comparison or alignment challenging. Naive fusion approaches often lead to misaligned predictions, particularly when one modality is ambiguous or incomplete. In this work, we propose an approach for OVDet that aligns relational structures across these incomparable spaces, ensuring optimal correspondence between visual and textual inputs. This shift from feature fusion to relational alignment bridges the gap between these spaces, enabling robust detection even when input from one modality is weak. Our evaluation on the challenging datasets demonstrates that our model sets a new benchmark in detecting rare objects, outperforming existing OVDet models. Additionally, we show that our multi-modal classifiers outperform single-modality models and even surpass fully-supervised detectors.

## 1 INTRODUCTION

In many real-world applications, such as e-commerce and autonomous systems, the range of objects a system needs to detect is constantly evolving. Traditional object detection models are limited by the fixed set of categories they were trained on, and when new products or object categories appear, these models require manual retraining, which is both costly and time-consuming Lin et al. (2014); Zhu et al. (2021); Redmon et al. (2016). Open-vocabulary object detection (OVDet) Zareian et al. (2021); Feng et al. (2022); Xu et al. (2024); Gu et al. (2022); Wang et al. (2024) addresses this limitation by enabling models to detect objects at inference time, without the need for retraining. Users can provide inputs through textual descriptions, image examples, or a combination of both, to identify objects of interest that were not explicitly part of the training data. This capability enables systems to adapt to new categories or unseen objects, offering the scalability required in dynamic environments. Existing OVDet approaches Zareian et al. (2021); Feng et al. (2022) address the challenge of detecting unseen objects by replacing the fixed classifiers in traditional detectors with text embeddings. These text embeddings are generated from pretrained text encoder using manual prompts, such as object class names or brief descriptions of the objects. While effective to some extent, these designs have notable limitations Lin et al. (2023); Wu et al. (2023); Kaul et al. (2023). *Lexical ambiguity*: some words have multiple meanings, and a simple text prompt cannot resolve these ambiguities. For example, "bat" can refer to both the animal and the sports tool, making it difficult for the model to interpret the correct meaning without additional context. *Lack of visual specificity*: text descriptions are often insufficient for conveying important visual details such as color, shape, or texture, which are essential for distinguishing between similar-looking objects. For example, describing different models of cars or species of animals requires detailed descriptions that are difficult to capture in simple prompts, whereas an image can provide all the necessary visual information instantly. *Unknown class names*: users may not always know the correct class name or how to describe the object they want to detect. In such cases, supplying an image example can bypass the need for an accurate verbal description.

To address these challenges, recent methods Wu et al. (2023); Lin et al. (2023) propose fusing visual and textual embeddings during inference to enhance object detection. The idea is to combine what the model sees in the image (visual data) with what it knows from text (descriptions or class names). However, these embeddings are learned from different modalities, each representing distinct types of

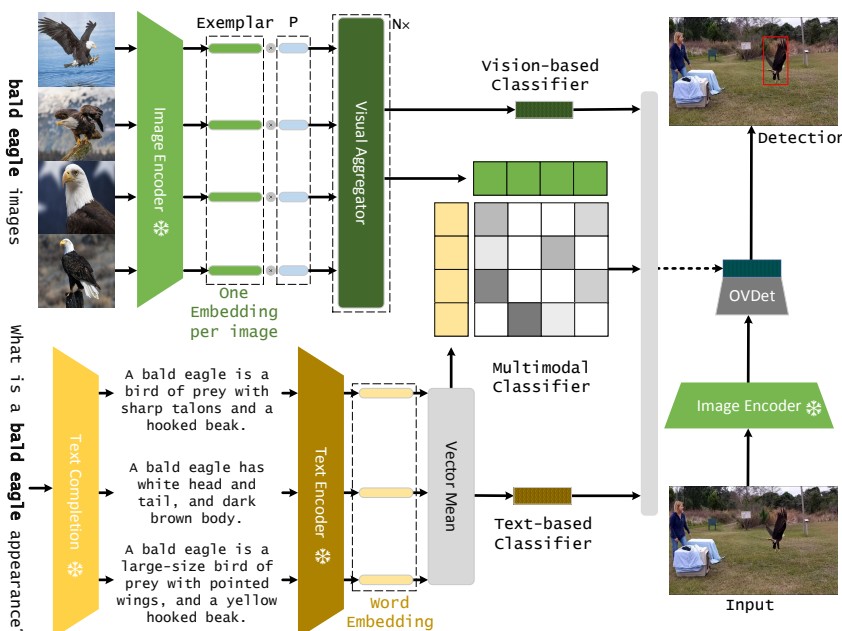

Figure 1: Overview of our model using text, vision, and multimodal classifiers for OVDet. Vision classifier (top-left) process $K$ examples per category through a frozen visual encoder, generating a refined embedding for each exemplar via a prototype discovery mechanism. These embeddings are then aggregated to form the final vision classifier. Text classifier (bottom-left) uses descriptive sentences generated by GPT-3, which are encoded by a text encoder. The resulting embeddings are averaged to construct the text classifier. Instead of a simple concatenation of features, our multimodal classifier (center) aligns both text and visual embeddings by leveraging feature-level and relational alignment, resulting in an improved combination of modalities for object detection.

information Ma et al. (2024b). A naive fusion assumes that these inputs are directly comparable and can be combined meaningfully, but in practice, the misalignment in their geometric and relational structures leads to poor generalization or incorrect object matching, especially for unseen categories.

We propose VOCAL (Vocabulary Alignment Classifier), a sophisticated approach to integrating visual and textual embeddings. Instead of relying on simple fusion methods, our approach aligns both feature-level and relational structures across the two modalities. By focusing on the contextual relationships between objects, our model finds the optimal mapping (correspondence) between visual and textual data. For instance, when a striped animal is described in text and an image of a zebra is provided as a visual example, our model aligns these inputs, even if one of them is unclear or incomplete. Rather than just matching individual objects, we capture how objects relate to one another in a broader context. This contextual understanding allows the model to infer the correct object, even when the input data is ambiguous. To further validate the effectiveness of this approach, we construct classifiers using either language descriptions or image examples and evaluate their impact individually. The proposed model is illustrated in Figure 1. Through a comprehensive evaluation on the challenging LVIS OVDet benchmark Gupta et al. (2019), we demonstrate several key advancements: by generating detailed language descriptions, we develop text-based classifiers that significantly outperform other methods that depend solely on class names. Using the image examples, we create vision-based classifiers capable of detecting new categories. We develop multimodal classifiers that outperform single-modality classifiers and achieve better results than existing methods.

## 2 RELATED WORK

**Closed-Vocabulary Object Detection.** Object detection has long been a cornerstone of computer vision, with a wide range of approaches developed over the years. Key methods can be broadly divided into one-stage and two-stage (or multi-stage) detectors. One-stage detectors, such as those

proposed by Redmon et al. (2016); Redmon & Farhadi (2018); Tan et al. (2020), perform classification and bounding box regression in a single step, often using predefined anchor boxes or directly detecting features like corners and center points. On the other hand, two-stage detectors first generate bounding boxes, then refine them into fixed-size region-of-interest (RoI) features for classification in the second stage Li et al. (2019); Cai & Vasconcelos (2018); Zhou et al. (2021). The use of Transformers Vaswani et al. (2017) in object detection, as proposed by Carion et al. Carion et al. (2020), marked a significant shift, treating object detection as a set prediction problem. Despite these advancements, traditional object detectors remain limited to recognizing only the objects present in their training datasets, lacking the ability to generalize to unseen classes during inference.

**Open-Vocabulary Object Detection (OVDet).** OVDet extends traditional object detection by allowing models to detect novel categories not present during training. To achieve this, OVDet leverages pretrained vision-language models (VLMs) like CLIP Radford et al. (2021) and ALIGN Jia et al. (2021), which are trained on large-scale image-caption pairs to associate visual features with natural language descriptions. For example, ViLD Gu et al. (2022) generates embeddings from image regions and matches them to object classes using a VLM, while RegionCLIP Zhong et al. (2022) employs region-text contrastive learning to recognize new objects. Other approaches like GLIP and MDETR Li et al. (2022); Kamath et al. (2021) align image and text features early on, framing detection as grounding textual descriptions within images. Zareian et al. Zareian et al. (2021) introduce OVR-CNN, which pretrains a visual encoder on image-caption pairs to build a comprehensive vocabulary. OWL-ViT Minderer et al. (2022) extends this by using larger transformer models and extensive image-caption datasets. OV-DETR Zang et al. (2022) adapts the DETR framework Carion et al. (2020) to handle open-vocabulary tasks. Detic and PromptDet Zhou et al. (2022); Feng et al. (2022) concurrently learn object localization and detailed vision-language matching by using max-size proposals to assign image-level labels. Recent methods Kaul et al. (2023); Ma et al. (2024b); Xu et al. (2024); Ren et al. (2023) fuse text and image embeddings, balancing uni-modal and multi-modal representations for better performance. CoDet Ma et al. (2024a) aligns object regions with textual descriptions based on their co-occurrence in large-scale image-text datasets, using contrastive learning to capture fine visual-language correlations. BARON Wu et al. (2023) adopts a bag-of-regions strategy, projecting contextually related regions into a word embedding space, aligned using contrastive learning. F-VLM Kuo et al. (2023) simplifies OVDet by leveraging frozen VLMs without knowledge distillation or weakly supervised learning. VLDet Lin et al. (2023) formulates region-word alignments as a set-matching problem and efficiently solves it using the Hungarian algorithm. By replacing the classification loss with a region-word alignment loss, VLDet improves novel category detection. DVDet Jin et al. (2024) introduces a visual prompt that refines region-text alignment by interacting with large language models to generate fine-grained descriptors. Our work builds on these advances, exploring various ways to construct classifiers that improve object detector generalization across diverse categories. Furthermore, recent works like those by Menon et al. Menon & Vondrick (2022), Pratt et al. Pratt et al. (2023), and Jin et al. Jin et al. (2024) employed GPT-3 Brown et al. (2020) to generate detailed class descriptions, enhancing zero-shot image classification. Our model similarly leverages natural language descriptions from large language models to enhance our textual classification for object detection.

## 3 METHOD

We propose VOCAL (Vocabulary Alignment Classifier) to detect and classify objects in images, including unseen categories. First, we provide an overview of OVDet (Section 3.1) followed by the construction of classifiers using language models (Section 3.2) and visual examples (Section 3.3). Finally, we explain the integration of these classifiers into a unified multimodal system in Section 3.4.

### 3.1 PRELIMINARY

In open-vocabulary object detection (OVDet), the input is an image $I \in \mathbb{R}^{3 \times H \times W}$, and the model produces two outputs: i) classification, which assigns a category label $c_j \in C_{\text{INF}}$ to each detected object $j$, where $C_{\text{INF}}$ represents the categories defined during inference; ii) localization, which predicts the bounding box coordinates $b_j \in \mathbb{R}^4$ indicating the precise position of each object within the image. Following Zareian et al. (2021); Zhou et al. (2022) our model is trained with two types of datasets. Specifically, a detection dataset $D_{\text{DET}}$ contains annotated images with bounding boxes and class labels covering a set of base categories $C_{\text{DET}}$. Image classification dataset $D_{\text{IMG}}$ consists of images

with class labels but no bounding boxes, covering a vocabulary $C_{\text{IMG}}$. The categories within $D_{\text{DET}}$ are known as base categories, whereas those appearing in $C_{\text{INF}}$ are identified as novel categories. Most OVDet models follow a multi-stage detection framework Zareian et al. (2021), comprising a visual encoder $\psi_{\text{EN}}$, a region proposal network $\psi_{\text{RP}}$, and an open-vocabulary classification module $\psi_{\text{CLS}}$. The process can be summarized as

$$\{c_j, b_j\}_{j=1}^M = \{\psi_{\text{bb}}(f_j), \psi_{\text{CLS}} \circ \psi_{\text{pro}}(f_j)\}_{j=1}^M$$
$$\{f_j\}_{j=1}^M = \psi_{\text{ROI}} \circ \psi_{\text{PG}} \circ \psi_{\text{EN}}(I) \tag{1}$$

The image $I$ is encoded into a set feature representation using an image encoder $\psi_{\text{EN}}$. The proposal generator $\psi_{\text{PG}}$ then identifies regions in the image that are likely to contain objects, and the pooling module $\psi_{\text{RP}}$ processes these proposals, generating feature vectors $\{f_j\}_{j=1}^M$, each corresponding to an object. The bounding box module $\psi_{\text{bb}}$ then predicts object positions $\{b_j\}_{j=1}^M$, while the classification module, consisting of a projection layer $\psi_{\text{pro}}$ and classifier $\psi_{\text{CLS}}$, assigns category labels $\{c_j\}_{j=1}^M$. In traditional closed-vocabulary settings, all components are trained jointly on $D_{\text{DET}}$. In OVDet, however, the classifiers $\psi_{\text{CLS}}$ are generated at inference time from external sources, such as pre-trained text encoders, enabling the model to adapt to novel categories $C_{\text{INF}}$ that differ from the training categories in $C_{\text{DET}}$. The following sections will explain how these classifiers are constructed.

### 3.2 TEXT-BASED CLASSIFIER WITH WEIGHTED CONTEXTUAL EMBEDDINGS

Traditional OVDet approaches, such as Detic Zhou et al. (2022) and ViLD Gu et al. (2022), rely on straightforward text-based classifiers generated from category names using simple prompts like "a photo of a(n) class name", which are then encoded using the CLIP text encoder. These methods often suffer from ambiguous representations, especially for categories with multiple meanings Wu et al. (2023). To address this, we enhance the generation of text-based classifiers by using a large language model like GPT-3 to generate multiple context-specific descriptions for each category $\{c_i\}_{i=1}^N$ ($N$ is the number of classes). We prompt the LLM with questions like "What does a $[c_i]$ look like?" or "Describe the visual characteristics of a $[c_i]$," generating five descriptions that capture different aspects of the object. However, not all descriptive elements are equally relevant to the visual features of the category. To address this, we introduce a weighted approach that focuses on selecting the most important elements from these descriptions. Given a set of $M$ descriptions $\{s_i^c\}_{i=1}^M$ for a class $c$, for each descriptive element $e_{ij}$, we calculate its relevance/alignment with the respective category's embedding. This is done by calculating the similarity between the element's embedding $(f_{\text{CLIP-T}}(e_{ij}))$ and the category's embedding $(f_{\text{CLIP-T}}(c))$. We then select the most relevant element $e_{max,i}^c$ from each $s_i^c$, which is the element with the highest similarity score $e_{max,i}^c = \arg\max_j s_{ij}^c$. This ensures that only the most relevant descriptive element is used to construct the classifier (the algorithm is given in 2.). The final classifier is constructed by averaging the embeddings of these relevant elements

$$w_{\text{TEXT}}^c = \frac{1}{M} \sum_{i=1}^M f_{\text{CLIP-T}}(e_{max,i}^c) \tag{2}$$

During training, these text-based classifiers are pre-computed for categories of interest in $C_{\text{DET}}$ and $C_{\text{IMG}}$, and are kept frozen throughout the training process. At inference, classifiers for unseen categories $C_{\text{INF}}$ are generated similarly, allowing the model to adapt to new categories effectively.

### 3.3 VISION-BASED CLASSIFIER WITH PROTOTYPE DISCOVERY

In addition to text-based classifiers, visual examples provide an alternative way to identify objects of interest at inference time. Visual examples are particularly effective for capturing fine-grained details that may be difficult to express in text, such as the complex wing patterns of a butterfly. For a given category $c$, let $\{x_i^c\}_{i=1}^K$ represent $K$ visual exemplars. These images are processed through a pre-trained CLIP visual encoder, resulting in embeddings $E_i^c = f_{\text{CLIP-IM}}(x_i^c)$, for $i = 1, 2, \ldots, K$. To capture the relationships between the image exemplars, we calculate a similarity matrix $S \in R^{K \times K}$, where its element $s_{ij}$ represents the similarity between the $i$-th and $j$-th image embeddings. A two-layer MLP (denoted as $\psi$) takes the similarity matrix $S$ as input and generates a probability vector $p \in R^K$, assigning probabilities to each exemplar, indicating how representative each one is for the category. Using these vectors, the prototype embedding $f_p^c$ for the category $c$ is computed

$$p = \text{softmax}(\psi(S)) \quad f_p^c = \sum_{i=1}^K p_i \cdot E_i^c \tag{3}$$

This prototype embedding focuses on the most representative features of the exemplars. To ensure consistency in the feature representation, each exemplar embedding $E_i^c$ is refined by blending it with the prototype embedding $f_p^c$. The new embedding is calculated as

$$\hat{E}_i^c = \lambda \cdot E_i^c + (1 - \lambda) \cdot f_p^c, \tag{4}$$

where, $\lambda$ controls the balance between the original embedding and the prototype, which set to 0.5 by default. Once the new embeddings are generated, they are passed through a multi-layer Transformer with a [CLS] token, and the output of the [CLS] token serves as the final vision classifier

$$w_{IMG}^c = \text{Transformer}(\{\hat{E}_i^c\}_{i=1}^K; t_{CLS}). \tag{5}$$

To enhance classifier discrimination, we employ contrastive learning with the InfoNCE loss, which pulls embeddings of the same category closer while pushing apart those of different categories. The model is trained offline using visual exemplars from large-scale datasets like ImageNet-21k Ridnik et al. (2021), which contains 11M images across 11,000 categories. During training, the CLIP visual encoder remains frozen to maintain consistency and ensure generalization to unseen categories Wu et al. (2023); Ma et al. (2024b). Once the model has been trained, the vision-based classifiers generated from the new embeddings are integrated into the overall OVDet model, and used during both training and testing phases, $C^{\text{DET}} \cup C^{\text{IMG}}/C^{\text{INF}}$. Our algorithm is described in Appendix 1.

### 3.4 MULTI-MODAL CLASSIFIER GENERATION

We extend the above methods by constructing classifiers that leverage the complementary strengths of text and image data. Text provides rich semantic relationships (e.g., dog and puppy), while images capture detailed spatial and appearance-based patterns. Directly combining these modalities is challenging due to differing feature representations Ma et al. (2024b). To address this, we propose an alignment mechanism that bridges the gap between text-based and vision-based classifiers. Given the visual embeddings $\{\hat{E}_i^c\}_{i=1}^K$ and from the image classifier and the text embeddings $\{s_j^c\}_{j=1}^M$ from the text classifier, we align these modalities in two steps: feature-level alignment and relational alignment. Let $A_{ij}$ be a degree of correspondence between the $i$-th visual embedding $\{\hat{E}_i^c\}_{i=1}^K$ and $j$-th text embedding $\{s_j^c\}_{j=1}^M$. The correspondence matrix $A \in R^{M \times K}$ helps minimize the distance between corresponding embeddings, $\sum_{i=1}^M \sum_{j=1}^K A_{ij} \|s_j^c - \hat{E}_i^c\|$. While feature-level alignment focuses on matching individual text and image embeddings, relational alignment is essential to ensure that the relationships between objects are preserved across both modalities. For example, text embeddings of lion and tiger are naturally close due to their semantic similarity, and this relationship should also be reflected in the visual embedding space. This alignment ensures that when the model encounters a novel category like a lion during inference, it can recognize it by relating it to a similar known category like tiger. To achieve this, we compute the pairwise relationships (distances) between text embeddings, represented as $R_{\text{TXT}} \in \mathbb{R}^{M \times M}$, and visual embeddings, represented as $R_{\text{IMG}} \in \mathbb{R}^{K \times K}$, and align them by minimizing the difference between distances across the two domains $\sum_{i,j,m,n} \left( R_{\text{TXT},ij}^c - R_{\text{IMG},mn}^c \right)^2 A_{im}^c A_{jn}^c$. Next, we combine this relational alignment with feature-level alignment (matching individual embeddings) into a single objective function

$$\alpha \cdot \sum_{i=1}^M \sum_{j=1}^K A_{ij}^c \|s_j^c - \hat{E}_i^c\|^2 + (1 - \alpha) \cdot \sum_{i,j,m,n} (R_{\text{TXT},ij}^c - R_{\text{IMG},mn}^c)^2 A_{im} A_{jn}, \tag{6}$$

where, $\alpha \in [0, 1]$ controls the balance between aligning individual features and maintaining relationships between embeddings. Once aligned, the final multi-modal classifier is constructed

$$w_{\text{MULTI}}^c = \sum_{i=1}^M \sum_{j=1}^K A_{ij}^c \left( s_j^c + \hat{E}_i^c \right). \tag{7}$$

This approach creates a robust and generalizable classifier, capable of identifying unseen categories in OVDet settings. Our algorithm is described in Appendix 3. Figure 1 presents a comprehensive pipeline highlighting our three classifiers.

## 4 EXPERIMENTS

**Benchmark setup.** We conduct our experiments using the LVIS benchmark Gupta et al. (2019), which contains annotations for 1203 classes across 100,000 images from MS-COCO. The dataset

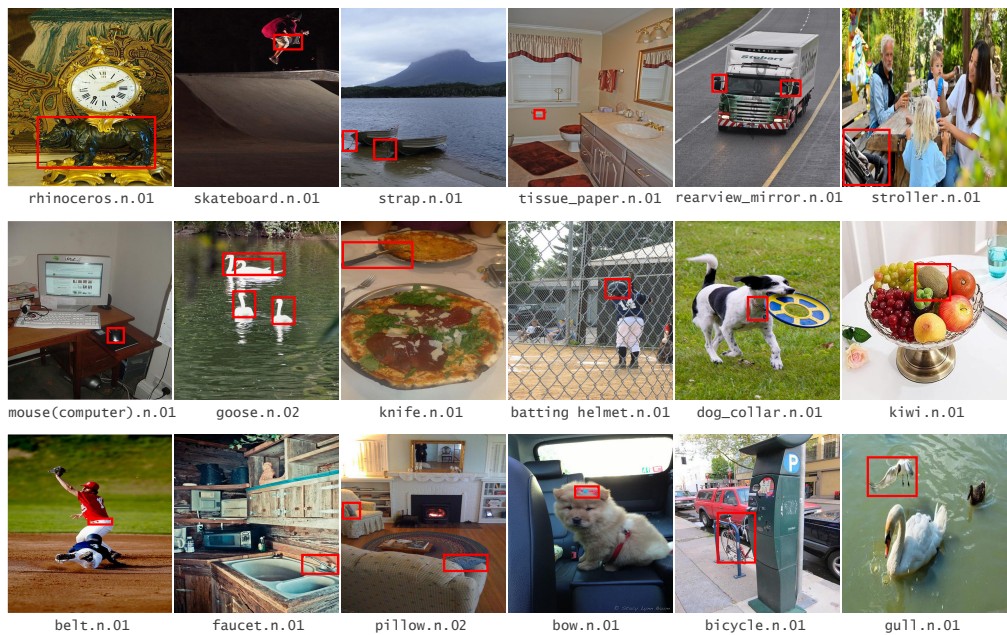

Figure 2: Qualitative examples of our model detecting rare categories in the LVIS validation set using text-based classifier. The classifier is generated from detailed descriptions provided by GPT-3.

provides bounding box and mask annotations for object instances, which are categorized as rare, common, and frequent, based on their occurrence in the dataset. To train our open-vocabulary object detector, we follow a setup similar to Zhou et al. (2022); Gu et al. (2022); Xu et al. (2024). Specifically, we use a filtered version of LVIS, where annotations for rare categories are removed, but the images containing these rare objects are kept. This reduced dataset, referred to as LVIS-filtered and denoted as $D_{\text{DET}}$, allows the model to learn from common and frequent categories while being evaluated on rare categories. Additionally, for image-level data ($D_{\text{IMG}}$), a subset of ImageNet-21K Deng et al. (2009) is used that overlap with the LVIS vocabulary. This subset is referred to as IN-LVIS, covering 997 of the 1203 classes in the LVIS dataset. The model's performance is evaluated on the LVIS validation set (LVIS-val), which includes all categories, but rare classes are treated as novel categories since no annotations for them were provided during training. We also conduct transfer experiments to show the generalization ability of our approach, evaluating our LVIS-trained model on the COCO Lin et al. (2014) and Objects365 Shao et al. (2019) validation sets. We report two evaluation metrics, Novel-AP and mAP. These metrics show that our model not only performs well on unseen categories (Novel AP) but also maintains strong overall performance (mAP).

**Implementation details.** For open-vocabulary LVIS experiments, we adopt CenterNet2 Zhou et al. (2021) with ResNet50 backbone He et al. (2016), pre-trained on ImageNet-21k-P Ridnik et al. (2021). The learning rate is warmed up to 2e-4 over the first 1000 iterations. The model is trained on the LVIS-filtered $D_{\text{DET}}$, for 90,000 iterations using Adam optimizer with batch size 64. When incorporating additional image-labeled data from ImageNet-21K (IN-LVIS), we perform joint training on both $D_{\text{DET}}$ and $D_{\text{IMG}}$, with a sampling ratio of 1:4. The batch size for this joint training is set to 64 for $D_{\text{DET}}$ and 256 for $D_{\text{IMG}}$, with image resolutions of $640 \times 640$ for $D_{\text{DET}}$ and $320 \times 320$ for $D_{\text{IMG}}$. We also set $\alpha = 0.5$, and $\lambda = 0.5$. All experiments are run on 4 NVIDIA 32GB GPUs.

**Constructing textual and visual classifiers.** For the textual classifier (Algorithm 1), we use GPT-3 from OpenAI to generate five descriptions for each class in the LVIS dataset. These descriptions are processed through the CLIP ViT-B/32 text encoder Radford et al. (2021), and the final token embedding from each input text is used to construct the classifier. To construct the vision-based classifier, we leverage CLIP ViT-B/32 as the visual encoder, pre-trained on ImageNet-21K-P Ridnik et al. (2021), a curated subset of ImageNet-21K containing around 11 million images from 11,000 classes. For each category, we use $K$ visual exemplars $\{x_i^c\}_{i=1}^{K}$, which are processed by the CLIP ViT-B/32 to produce visual embeddings. We apply adaptive image augmentation (AIA), augmenting

Table 1: Open-vocabulary detection performance on LVIS. Rows for our models are highlighted in green and yellow, representing results from text, vision, and multimodal classifiers. Models are divided into those trained only on LVIS-filtered (top) and those incorporating additional images (bottom). Due to computing limitations, we compare to models which use ResNet-50 He et al. (2016) or similar architectures.

| Method | Detector backbone | Extra data | Novel AP | AP |
|---|---|---|---|---|
| Detic Zhou et al. (2022) | RNet-50 | - | 16.5 | 30.0 |
| PromptDet Feng et al. (2022) | RNet-50 | - | 19.0 | 21.4 |
| OVDETR Zang et al. (2022) | DETR+RNet-50 | - | 17.4 | 26.6 |
| DetPro Du et al. (2022) | RNet-50 | - | 19.8 | 25.9 |
| ViLD Gu et al. (2022) | RNet-50 | - | 16.6 | 25.5 |
| MMOVD Kaul et al. (2023) | RNet-50 | - | 19.3 | 30.6 |
| BARON Wu et al. (2023) | RNet-50 | - | 19.2 | 26.5 |
| F-VLM Kuo et al. (2023) | RNet-50 | - | 18.6 | 24.2 |
| DVDet Jin et al. (2024) | RNet-50 | - | 21.3 | 28.1 |
| VLDet Lin et al. (2023) | RNet-50 | - | 21.7 | 30.1 |
| OVMR Ma et al. (2024b) | RNet-50 | - | 21.2 | 30.0 |
| OVMR- T Ma et al. (2024b) | RNet-50 | - | 19.0 | 29.6 |
| VOCAL- T | RNet-50 | - | 21.7 | 30.3 |
| VOCAL- V | RNet-50 | - | 21.2 | 29.7 |
| VOCAL- MM | RNet-50 | - | **22.8** | 30.8 |
| OWL-ViT Minderer et al. (2022) | ViT-B/32 | LiT | 19.7 | 23.5 |
| RegionCLIP Zhong et al. (2022) | RNet-50 | CC3M | 17.3 | 28.3 |
| PromptDet Feng et al. (2022) | RNet-50 | LAION | 21.4 | 25.3 |
| Detic Zhou et al. (2022) | RNet-50 | IN-LVIS | 24.6 | 32.5 |
| POMP Ren et al. (2023) | ViT-B/32 | IN-LVIS | 26.8 | 36.2 |
| CoDet Ma et al. (2024a) | RNet-50 | CC3M | 23.4 | 30.7 |
| VOCAL- T | RNet-50 | IN-LVIS | 26.9 | 33.0 |
| VOCAL- V | RNet-50 | IN-LVIS | 25.1 | 31.6 |
| VOCAL- MM | RNet-50 | IN-LVIS | **28.5** | 33.7 |
| Fully-Supervised Zhou et al. (2022) | RNet-50 | - | 25.5 | 31.1 |

each exemplar five times before passing them through the CLIP encoder, resulting in $5K$ augmented visual embeddings per class. These augmented embeddings are refined using our prototype discovery method (as described in 3.3), which ensures that the most representative features are aggregated into the final classifier. The refined embeddings are then processed through 4 transformer blocks, each with an output dimension of 512, and an MLP with a dimension of 2048. These blocks aggregate the refined embeddings into a cohesive classifier representation. The vision-based classifier is trained using visual exemplars from the ImageNet-21K-P. LVIS-filtered data is used to train the open-vocabulary object detector, and IN-LVIS serves as an additional source of weak supervision. Figure 2 shows an example of our model detecting the rare categories from the LVIS validation set.

**Multi-modal classifier generation.** To construct the multi-modal classifier, we combine both text-based and vision-based classifiers to capture complementary information from both modalities. Text embeddings are generated from category descriptions, while vision embeddings are generated from augmented visual exemplars. These embeddings are aligned at both the feature level and the relational level, and the final multi-modal classifier is built by aggregating the aligned embeddings from both modalities, allowing the model to effectively handle open-vocabulary object detection tasks. Additionally, for comparison, we test the effectiveness of our visual classifier by combining our text-based classifiers with the baseline vision-based classifiers, as described in the ablation study.

### 4.1 MAIN RESULTS

**Open-Vocabulary LVIS benchmark.** Table 1 shows the performance comparisons on the open-vocabulary LVIS object detection using Novel AP (for rare categories) and AP (for overall per-

Table 2: Cross-datasets transfer detection from LVIS to COCO and Objects365.

| Method | Target Dataset: COCO | | | Target Dataset: Objects365 | | |
|---|---|---|---|---|---|---|
| | AP | AP-50 | AP-75 | AP | AP-50 | AP-75 |
| DetPro Du et al. (2022) | 34.9 | 53.8 | 37.4 | 12.1 | 18.8 | 12.9 |
| ViLD Gu et al. (2022) | 36.6 | 55.6 | 39.8 | 11.8 | 18.2 | 12.6 |
| Detic Zhou et al. (2022) | 38.8 | 56.0 | 41.9 | 13.9 | 19.7 | 15.0 |
| F-VLM Kuo et al. (2023) | 32.5 | 53.1 | 34.6 | 11.9 | 19.2 | 12.6 |
| BARON Wu et al. (2023) | 36.6 | 55.7 | 39.1 | 13.6 | **21.0** | 14.5 |
| CoDet Ma et al. (2024a) | 39.1 | 57.0 | 42.3 | 14.2 | 20.5 | 15.3 |
| VOCAl (Ours) | **40.3** | **57.9** | **43.5** | **15.0** | 20.7 | **16.1** |

Table 3: Ablation study evaluating the performance of our vision-based, text-based, and multimodal classifiers on the LVIS OVDet benchmark. Vision-based classifiers ( red rows) are compared to a baseline that uses a simple mean of visual embeddings (V-Mean), demonstrating the effectiveness of our prototypical embedding strategy. Multimodal classifiers ( orange rows) outperform both vision-only and text-only classifiers ( gray row), emphasizing the advantage of combining visual and textual information for detecting rare and unseen categories. The left half of the table shows results from models trained on LVIS-filtered, while the right half incorporates extra image data (LVIS-filtered + IN-LVIS), illustrating how additional data further enhances performance.

| Model | V-CLS | V-Mean | T-CLS | Extra | APr | AP | Model | V-CLS | V-Mean | T-CLS | Extra | APr | AP |
|---|---|---|---|---|---|---|---|---|---|---|---|---|---|
| X-A | ✓ | | | | 21.2 | 29.7 | X-F | ✓ | | | ✓ | 25.1 | 31.6 |
| X-B | | ✓ | | | 17.6 | 28.5 | X-G | | ✓ | | ✓ | 22.7 | 31.2 |
| X-C | | | ✓ | | 21.7 | 30.3 | X-H | | | ✓ | ✓ | 26.9 | 33.0 |
| X-D | ✓ | | ✓ | | 22.8 | 30.8 | X-I | ✓ | | ✓ | ✓ | 28.5 | 33.7 |
| X-E | | ✓ | ✓ | | 21.6 | 39.5 | X-J | | ✓ | ✓ | ✓ | 27.9 | 33.2 |

formance). In OVDet, Novel AP is critical, as it measures the model's ability to detect unseen objects. In the LVIS-filtered setup, where no additional image data is used, our multi-modal model (VOCAL-MM) achieves a Novel AP of 22.8, establishing new benchmarks in detecting unseen and rare categories. This marks a +1.1 improvement over VLDet (21.7) and +1.6 over OVMR (21.2). Our text-based (VOCAL-T) and vision-based (VOCAL-V) classifiers also demonstrate strong results with Novel APs of 21.7 and 21.2, respectively. When incorporating additional image-level data, our results are even more striking, with a 28.5 Novel AP, outperforming Detic by +16% and PromptDet by +33% in Novel AP. The standout performance of our models, especially in detecting rare and unseen categories is attributed to the seamless integration of textual and visual information. The alignment between two complementary modalities at both the feature level and relational level ensures that the classifier captures not just the visual appearance of objects, but also their semantic context, leading to superior performance in open-vocabulary detection tasks. Some works like RO-ViT and DITO use larger backbones (e.g., Swin-B/L Liu et al. (2021)), but due to limited computational resources, we focus on comparisons with models using ResNet-50 He et al. (2016) backbones or similar.

**Transfer to other datasets.** We evaluate our model's ability to generalize across different domains using cross-dataset transfer detection, where the detector trained on LVIS is applied to COCO and Objects365 without fine-tuning. As shown in Table 2, among the open-vocabulary models, our approach achieves the strongest transfer performance, an AP of 40.3/15.0 on COCO/Objects365, outperforming CoDet by +1.2/+0.8, and BARON by +3.7/+1.4. These results highlight the robustness and generalization ability of our model in handling object detection tasks across diverse domains.

**Ablation study.** In OVDet methods, the focus is often on text-based classifiers, with vision-based classifiers receiving less attention. To address this gap, we compare our proposed vision-based classifier, as detailed in Section 3.3, to a baseline classifier that uses a straightforward mean of visual embeddings generated by the CLIP visual encoder ($\frac{1}{K} \sum_{i=1}^{K} E_i^c$), and does not incorporate our prototype discovery strategy. The red rows in Table 3 highlight the comparison between our complete vision classifier and its baseline. When trained without additional image data (left-half of the table),

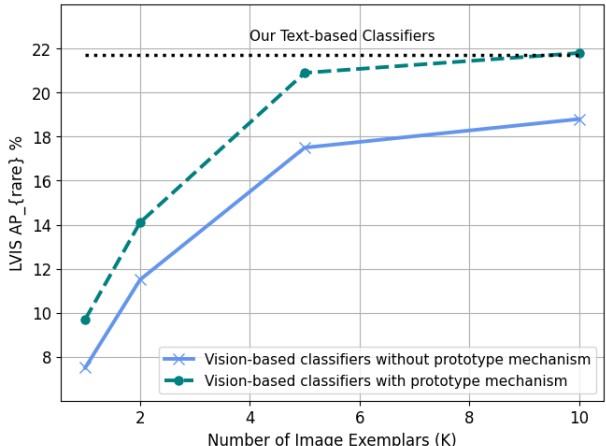

Figure 3: Vision-based classifiers using different numbers of image exemplars per class ($K = 1, 2, \ldots, 10$), on the LVIS OVOD. Optimal performance is achieved with $K = 5$.

our refined vision classifier (X-A) achieves a +3.6 APr improvement for rare categories over the baseline model (X-B). When extra data is used (IN-LVIS), our model (X-F) further outperforms its baseline counterpart (X-G) by +2.4 APr. These results demonstrate the effectiveness of our prototypical embedding strategy in constructing effective vision-based classifiers, as opposed to simply averaging the visual exemplars. All results were based on $K = 5$. Similarly, the orange rows in Table 3 show the performance of our multimodal (MM) classifiers. Without additional image-level data, the MM classifier (X-D) achieves a +1.2 APr gain over its baseline (X-E). When using additional data, we see a smaller improvement of +0.6. We also note that for constructing our MM classifier, as detailed in (3.4), we used the description embeddings $\{s_j\}_{j=1}^M$. Interestingly, when comparing the use of raw text embeddings to the refined $e_{max}$, the raw descriptions result in a +0.8 gain with extra data, whereas $e_{max}$ provides a +0.6 gain in scenarios without additional data. Lastly, by comparing the text-based classifier (gray row) with the multimodal classifiers (yellow rows), we observe that in all cases, adding visual examples improves performance. This clearly demonstrates that the combination of visual and text embeddings in multimodal classifiers significantly boosts performance, particularly in detecting unseen categories.

## 5 USING IMAGE EXEMPLARS

This section presents the results of different numbers of $K$ image exemplars per class used for our visual classifiers. Figure 3 illustrates the detection results on the LVIS OVOD benchmark for rare categories with $K = \{1, 2, \ldots 10\}$. We compare our method, which incorporates prototype embeddings (green dashed line), against a simple vector mean of the embeddings (blue line) for the $K$ exemplars. Across all values of $K$, our classifier consistently improves performance on rare classes, demonstrating its ability to effectively extract and combine the most relevant information from the exemplars. The optimal performance is achieved with $K = 5$, and even for $K = 1$, our model provides a +2.5 APr boost over the baseline.

## 6 CONCLUSION

In this paper, we address the challenges of open-vocabulary object detection (OVDet) by focusing on the integration of text and image data to generate robust classifiers. Unlike other methods that rely on simple class names, our approach leverages large language models to generate rich, context-aware descriptions for each object category. We further enhance the detection capabilities by incorporating visual exemplars, enabling our model to capture fine-grained visual details that are often difficult to express in text. By aligning the feature and relational structures between text and image embeddings, our method achieves a more accurate and flexible detection framework. The resulting classifiers outperform existing approaches in identifying unseen categories, pushing the boundaries of OVDet.

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

## A  APPENDIX

The proposed algorithms for vision, text, and multimodal classifiers.

---

**Algorithm 1** Text-based Classifier with Weighted Contextual Embeddings

---

**Require:** $C$: Set of categories $\{c_i\}_{i=1}^N$, $f_{\text{CLIP-T}}$: Pre-trained CLIP text encoder, $M$: Number of descriptions per category, LLM: Large language model (e.g., GPT-3)

**Ensure:** $w_{\text{TEXT}}^c$: Text-based classifier for category $c$

1: **Step 1: Generate Descriptions**
2: **for** each category $c$ **do**
3:     $\{s_i^c\}_{i=1}^M \leftarrow$ LLM(Prompts for category $c$)      ▷ Generate $M$ descriptions per category using the LLM
4: **end for**
5: **Step 2: Compute Element Similarities**
6: **for** each category $c$ **do**
7:     **for** each description $s_i^c$ **do**
8:         **for** each descriptive element $e_{ij}$ in $s_i^c$ **do**
9:             $E_{ij}^c \leftarrow f_{\text{CLIP-T}}(e_{ij})$                     ▷ Compute embedding of descriptive element
10:             $s_{ij}^c \leftarrow \cos(E_{ij}^c, f_{\text{CLIP-T}}(c))$     ▷ Calculate similarity between element and category embedding
11:         **end for**
12:         $e_{max,i}^c \leftarrow \arg\max_j s_{ij}^c$          ▷ Select the most relevant element with highest similarity score
13:     **end for**
14:     **Step 3: Construct Classifier**
15:     $w_{\text{TEXT}}^c \leftarrow \frac{1}{M}\sum_{i=1}^M f_{\text{CLIP-T}}(e_{max,i}^c)$                ▷ Average embeddings of the most relevant elements
16: **end for**
17: **return** $w_{\text{TEXT}}^c$                     ▷ Return the text-based classifier for each category

---

---

**Algorithm 2** Vision-based Classifier with Prototype Discovery

---

**Require:** $\{x_i^c\}_{i=1}^K$: Visual exemplars for category $c$, $f_{\text{CLIP-IM}}$: Pre-trained (Frozen) CLIP visual encoder,, $\psi$: Two-layer MLP, $S$: Similarity matrix, $t_{\text{CLS}}$: [CLS] token, $\lambda$

**Ensure:** $w_{\text{IMG}}^c$: Vision-based classifier for category $c$

1: **Step 1: Embedding Extraction**
2: **for** each exemplar $x_i^c$ **do**
3:     $E_i^c \leftarrow f_{\text{CLIP-IM}}(x_i^c)$                  ▷ Extract embeddings
4: **end for**
5: **Step 2: Similarity Matrix Calculation**
6: $S[i,j] \leftarrow \cos(E_i^c, E_j^c)$           ▷ Compute similarity between exemplar embeddings
7: **Step 3: Prototype Discovery**
8: $p \leftarrow \text{softmax}(\psi(S))$            ▷ Process the similarity matrix through MLP $\psi$
9: $f_p^c \leftarrow \sum_{i=1}^K p_i \cdot E_i^c$            ▷ Compute prototype embedding for category $c$
10: **Step 4: Adaptive Refinement**
11: **for** each exemplar embedding $E_i^c$ **do**
12:     $\hat{E}_i^c \leftarrow \lambda_i \cdot E_i^c + (1 - \lambda_i) \cdot f_p^c$         ▷ Refine embedding using prototype $f_p^c$
13: **end for**
14: **Step 5: Vision Classifier**
15: $w_{\text{IMG}}^c \leftarrow \text{Transformer}(\{\hat{E}_i^c\}_{i=1}^K, t_{\text{CLS}})$    ▷ Generate classifier with [CLS] token from Transformer
16: **Step 6: Contrastive Learning**
17: Apply contrastive learning with InfoNCE loss to improve discrimination:
18: **return** $w_{\text{IMG}}^c$

---

**Algorithm 3** Multi-modal Classifier Generation with Feature and Relational Alignment

---

**Require:** $\{\hat{E}_i^c\}_{i=1}^K$: Visual embeddings, $\{s_j^c\}_{j=1}^M$: Text embeddings, $\alpha$

**Ensure:** $w_{\text{MULTI}}^c$: Multi-modal classifier for category $c$

1: **Step 1: Feature-level Alignment**
2: Compute the correspondence matrix $A_{ij}$       ▷ Align individual text and image embeddings (sec3.4)
3: **Step 2: Relational Alignment**
4: Compute $R_{\text{TXT}}$ and $R_{\text{IMG}}$                     ▷ Refer to sec3.4
5: Minimize the difference between text and image embeddings ▷ Ensure relationships between text and visual embeddings are consistent
6: **Step 3: Joint Objective**
7: Combine feature and relational alignment                 ▷ Refer to Eq. 6
8: **Step 4: Construct Multi-modal Classifier**
9: Combine aligned text and visual embeddings           ▷ Refer to Eq. 7
10: **return** $w_{\text{MULTI}}^c$

---

