# OpenReview forum: "Open-Vocabulary Object Detection for Incomparable Spaces"
_ICLR.cc/2025/Conference — ICLR 2025 Conference Withdrawn Submission_

### Official Review · Reviewer_3vd5 · 2024-11-02

**Soundness:** 2
**Presentation:** 2
**Contribution:** 2
**Rating:** 5
**Confidence:** 4

**Summary:**

This paper introduces a novel open-vocabulary detection method that utilizes both textual and visual classifiers, integrating them through feature-level alignment and relational alignment. The author conducts experiments on LVIS to demonstrate its performance on novel categories.

**Strengths:**

1. The feature-level alignment and relational alignment for fusing textual and visual classifiers is very interesting.
2. The weighted contextual embeddings and prototype discovery respectively optimize the methods for constructing textual and visual classifiers.

**Weaknesses:**

1.	The method’s pipeline is similar to MMOVD[1], with only minor improvements made to the construction and fusion of classifiers. Overall, the novelty might be quite limited.
2.	The method is similar to MMOVD, but lacks critical experiments comparing it with MMOVD, such as evaluations using IN-LVIS as extra data on the LVIS dataset, and MMOVD’s evaluations on cross-dataset transfer detection.
3.	There are missing experiments that prove the effectiveness of the method. 1) Lack of experiments demonstrating that weighted contextual embeddings improve the performance of a text-based classifier compared to simply averaging; 2) Lack of experiments showing that using feature-level alignment and relational alignment is more effective compared to naive fusion strategies like addition.
4.	The comparison experiments between V-CLS and V-Mean are not reasonable. V-CLS, compared to V-Mean, uses both the prototype discovery strategy and additional transformer blocks as the Visual Aggregator. This setup does not validate the effectiveness of the prototype discovery strategy. According to MMOVD[1], using a Visual Aggregator already performs better than directly averaging various visual embeddings. V-CLS should be compared with a Visual Aggregator that does not use the prototype discovery strategy.
5.	There is a lack of hyperparameter analysis for $\lambda$ and $\alpha$.
6.	Results of open vocabulary object detection evaluations on the COCO dataset are missing.

[1] Multi-Modal Classifiers for Open-Vocabulary Object Detection, ICML 2023

**Questions:**

1. Figure 2 attempts to demonstrate the model’s effectiveness in detecting rare categories, but the examples provided belong to either the frequent or common categories, which does not prove the model’s capability in detecting rare categories. For instance, ‘knife’, ‘skateboard’, ‘belt’, ‘pillow’, and ‘bicycle’ are all frequent categories, while ‘rhinoceros’, ‘goose’, ‘kiwi’, and ‘gull’ belong to common categories.
2. Please refer to the weakness section.

---

### Official Review · Reviewer_Jt2n · 2024-11-02

**Soundness:** 3
**Presentation:** 3
**Contribution:** 3
**Rating:** 6
**Confidence:** 3

**Summary:**

The paper introduce an approach for open-vocabulary object detection (OVDet) that aligns relational structures across visual and textual data to enhance the detection of objects, especially unseen or rare objects. The authors propose a model called VOCAL (Vocabulary Alignment Classifier) that shifts from feature fusion to relational alignment, bridging the gap between visual and textual inputs. VOCAL leverages both text descriptions and image examples to identify objects, addressing limitations such as lexical ambiguity, lack of visual specificity, and unknown class names. The evaluation on challenging datasets shows that VOCAL outperforms existing OVDet models and even surpasses fully-supervised detectors in detecting rare objects.

**Strengths:**

1. The paper presents a sophisticated method for OVDet that focuses on relational alignment between visual and textual data, which is a novel approach in the field.

2. VOCAL demonstrates superior performance in detecting rare objects and outperforms existing OVDet models, which is a significant achievement.

3. The model demonstrates a new benchmark in detecting rare objects and outperforms existing OVDet models, which is a substantial achievement.

**Weaknesses:**

1.  The approach may be more complex and computationally intensive than simpler fusion methods, which could be a limitation in resource-constrained environments.

2. The introduction of the Image and Text Encoder results in a detection process that requires more computation, and fairness compared to other OVDet methods needs to be considered.

3. Some related OVDet methods are missing. For example, Distilling DETR with Visual-Linguistic Knowledge for Open-Vocabulary Object Detection ICCV 2023.

**Questions:**

See Weaknesses. My major concern is introducing much more complexity compared with previous methods.

---

### Official Review · Reviewer_JSDD · 2024-11-03

**Soundness:** 2
**Presentation:** 2
**Contribution:** 2
**Rating:** 5
**Confidence:** 3

**Summary:**

This paper addresses the challenges of open-vocabulary object detection (OVDet), where the goal is to detect objects at inference time that were not seen during training. The authors propose an approach called VOCAL (Vocabulary Alignment Classifier), which integrates visual and textual embeddings by aligning both feature-level and relational structures across these two modalities. This method aims to bridge the gap between visual and textual data, enabling robust detection even when input from one modality is weak or ambiguous.

**Strengths:**

1. It combines textual descriptions and visual examples to identify objects, leveraging the strengths of both modalities to improve detection accuracy.
2. Instead of simple feature fusion, VOCAL focuses on aligning the contextual relationships between objects in text and images, which is a novel way to handle the misalignment problem in heterogeneous data. The model can adapt to new categories or unseen objects without retraining, which is a significant advantage in dynamic environments where new objects frequently appear.
3. The evaluation shows that the model outperforms existing OVDet models, setting new benchmarks in detecting rare objects.

**Weaknesses:**

1. The method involves complex alignment mechanisms that could be computationally expensive and may require substantial resources for training and inference.
2. The performance of VOCAL heavily relies on the quality of the text and image embeddings. If the embeddings are not representative, the alignment may not be effective.
3. While the model can adapt to new categories, the scalability to a very large number of categories or extremely rare objects is not explicitly discussed and could be a challenge.
4. Although the paper mentions cross-dataset transfer, the generalization of the model to datasets outside of the trained domain is a potential concern that may require further validation.

**Questions:**

Pls see the weeknesses above.

---

### Note · Authors · 2024-12-01

I have read and agree with the venue's withdrawal policy on behalf of myself and my co-authors.